# Are People with Aphasia (PWA) Involved in the Creation of Quality of Life and Aphasia Impact-Related Questionnaires? A Scoping Review

**DOI:** 10.3390/brainsci10100688

**Published:** 2020-09-29

**Authors:** Marina Charalambous, Maria Kambanaros, Jean-Marie Annoni

**Affiliations:** 1Laboratory of Cognitive and Neurological Sciences, University of Fribourg, CH-1700 Fribourg, Switzerland; jean-marie.annoni@unifr.ch; 2Allied Health and Human Performance, University of South Australia, Adelaide SA 5001, Australia; maria.kambanaros@unisa.edu.au

**Keywords:** people with aphasia (PWA), stroke, patient and public involvement (PPI), inclusion, communication impairment

## Abstract

Background: Quality of Life (QoL) questionnaires are used to describe the impact of aphasia on stroke survivors’ life. People with aphasia (PWA) are traditionally excluded from research, potentially leading to a mismatch between the factors chosen in the tools and the realistic needs of PWA. The purpose of this review was to determine the direct involvement of PWA in the creation of QoL and aphasia impact-related questionnaires (AIR-Qs). Methods: A scoping review methodology was conducted by an expert librarian and two independent reviewers on health sciences based on the Preferred Reporting Items for Systematic Reviews and Metanalyses extension for Scoping Reviews (PRISMA-ScR) protocol, through a literature search in five databases: Medline Complete, PubMed, PsychINFO, Scopus, and Google Scholar. Search terms included ‘stroke’, ‘people with aphasia’, ‘communication’, ‘well-being’, and ‘quality of life’. Results: Of 952 results, 20 studies met the eligibility criteria. Of these, only four AIR-Qs studies (20%) were found reporting the direct involvement of PWA, while no QoL tools did so. Evidence showed involvement in the creation phase of AIR-Q, mainly in a consultation role. Conclusions: There is an absence of a framework for conducting and reporting the involvement of PWA in qualitative participatory research studies, which limits effectiveness to promote equitable best practice in aphasia rehabilitation.

## 1. Introduction

Research and medical attention are primarily focused on the areas of primary and secondary prevention, acute management, and early rehabilitation of stroke. The neurological sequelae of stroke bring about many issues for the stroke survivor to deal with, such as maintaining relationships, issues with self-confidence, managing finances, cognitive disorders, and communication difficulties due to aphasia. Aphasia affects approximately 20% of chronic stroke survivors and impacts on one or more areas of communication such as the ability to speak, understand, read, and write [1]. Aphasia is linked to poorer functional recovery, return to work, and activities of daily living and leads to fewer friendships, smaller social networks, and reduced quality of life (QoL) [2,3,4,5].

Yet people with aphasia (PWA) after stroke, on discharge, from a largely medicalized pathway, enter a world where support is usually unclearly defined, often fragmented or non-existent, which dramatically decrease QoL. There is a large body of literature on the use of questionnaires to describe the impact of aphasia on stroke survivors’ QoL. The findings demonstrate robust evidence for the severe effects of communication deficits on social integration and well-being for PWA post-stroke. Health professional researchers have developed QoL-type of questionnaires within the International Classification Functioning (ICF) framework [6] to examine these issues [7]. PWA are traditionally excluded from participating in research, because of perceived communication difficulties and assumptions about inaccurate responses [8,9,10]. As a result, there is potentially a mismatch between the factors chosen in the tools by researchers and the realistic needs of PWA. Inconsistent or no involvement of PWA in the creations of these tools creates a fragmented evidence base, making it difficult to draw conclusions on what is important for whom, why, and in what context. In existing scientific publications, there is insufficient reporting on the frequency and level of involvement of PWA as co-creators in conceiving QoL and aphasia impact-related questionnaires (AIR-Qs).

There is an increasing drive among patients, clinicians, patient advocates (support organizations and medical associations), politicians, and researchers toward the engagement of people who directly experience a condition and other layperson/nonprofessional service users (i.e., carers, family members, support group members) in health-related qualitative research [11]. Advocates of this framework argue that patient stakeholders (patients, families, and caregivers) have a deeper knowledge of the related condition, gained through living with and managing the illness on a daily basis, knowledge that is usually ignored both by clinicians and researchers [12,13,14].

In the past decade, researchers in healthcare have shifted their attention to issues closely related to the patient’s needs and desires, by engaging patients as co-researchers and research partners in studies through the patient and public involvement (PPI) approach as opposed to passive study participants [15]. According to the National Institute of Health Research (NIHR) in the UK, PPI is the active partnership between patients, the public and researchers in the research process, as opposed to the role of people as ‘subjects’ of research. PPI is defined as carrying out research ‘with’ or ‘by’ people who use services rather than ‘to’, ‘about’, or ‘for’ them [16]. The INVOLVE Organization in the UK [16] states that PPI would include contribution in the choice of research topics, assisting in the study design, advising on the research project or in implementing the research, interpretation of results, and dissemination. The PPI evidence base has expanded significantly over the past decade in health sciences, facilitated by Staniszewska and colleagues [15] in the development of the Guidance for Reporting Involvement of Patients and Public (GRIPP) checklist, which serves as a framework for reporting and involving patients and the public in research. Nevertheless, the reporting of PPI in published papers related to the QoL of PWA has often been inconsistent or partial. There is little or no information about the context, the process, and the impact of PPI in stroke aphasia research, coupled with limited reporting on the conceptualization or the theoretical underpinning regarding the involvement of PWA as stakeholders.

In the literature the term public and patient ‘involvement’ is used interchangeably or synonymously with terms such as: Patient engagement, layperson, PPI contributor, peer research, expert by experience, consumer, service user, stakeholders, stakeholder engagement, user involvement, research partners, patient partner, and co-researcher [17]. There are different levels of involvement according to the conceptual framework proposed by the Irish Health Research Forum(IHRF). The IHRF framework [18] states that patients could be involved in PPI research at various levels starting with (1) providing basic information about their condition, (2) having a consultation role throughout the research process, (3) taking an active role in research planning and decision making, (4) initiating research and being actively involved throughout the process and, finally, (5) having full control of the study and work in partnership with the research team from passive partners to active leading roles. Patient involvement ranges from stakeholders’ input, to consultation, to collaboration, or shared leadership [19,20]. PPI can be incorporated in ad hoc working groups to develop dissemination strategies or to provide input in an advisory committee or co-researcher capacity [18].

According to the framework proposed [11] for implementing PPI in research, there are three phases in Patient and Service User Engagement (PSUE), as follows:(1)Preparatory Phase: Setting the agenda and determining the funding resources for the research. Prioritize key topics and questions and work on research preparation such as establishing a steering committee, prepare protocols, reviews, etc.(2)Execution Phase: Work on study design and procedure, i.e., preparation of consent procedures, development of outcome instruments, study recruitment, data collection, and data analysis.(3)Translational Phase: Work on dissemination, i.e., development of manuscripts, pamphlets, social media campaign, etc.; implementation of the study, i.e., developing clinical practice guidelines and evaluation, i.e., evaluation of the research process; and future research ideas.

Still, the PPI concept remains problematic, as it does not cover the pragmatic issues and complexities of involving patients with communication impairment and/or other disabilities in the research process. There is still a lack of clarity in the selection process of PPI contributors, and researchers do not always feel ready to support PPI roles [21]. There is a shortcoming in role expectations from all stakeholders and PPI research should offer flexibility in terms of level of involvement by co-creating accessible research processes [18]. This scoping review aimed to explore the evidence gap in the literature on the involvement of PWA as co-researchers, stakeholders, and patient partners in the creation of QoL and AIR-Qs.

Objective of the Scoping Review. The aim of this review was to determine the evidence showing the direct involvement of PWA in the creation of QoL and AIR-Qs. The purpose was to explore the presence versus the absence of the contribution of PWA as research partners in the creation of QoL and AIR-Qs. The main research questions were five-fold: (1) To investigate whether researchers included PWA in the creation of the questionnaires, (2) to document the type of involvement of PWA when building the questionnaires, (3) to establish if these tools were pilot tested on PWA before publication, (4) to examine the presence of the term ‘patient involvement’ in the published research, and (5) to determine any additional issues PWA would add to the creation of QoL and/or AIR-Qs.

## 2. Materials and Methods

To address the involvement of PWA in the creation of QoL and AIR-Qs, a scoping review was conducted. This methodology is found to be an effective approach for examining research topics with developing evidence that is not widely reviewed [22,23]. The scoping review methodology allows researchers to include broad questions and a wide range of search approaches around a specific topic of interest promoting the direction of future research in the proposed field [24]. The current scoping review included a systematic search of the literature related to QoL and AIR-Qs for PWA after stroke.

The scoping review was implemented based on the methodological framework of Arksey and O’Malley [22] for conducting a scoping study. The framework consists of five steps: (1) Identifying the research questions, (2) identifying relevant studies, (3) selecting studies, (4) charting the data, and (5) collating, summarizing, and reporting the results.

The research questions and the search terms were developed in partnership among the researchers (authors). The search terms were related to the targeted population, the proposed tools, and the types of study design to include in the review. An expert librarian on health sciences initiated a literature search in April 2020. Search strings were based on Mesh terms. The string of keywords searched were: Stroke OR cerebrovascular accident OR cva AND aphasia OR dysphasia OR aphasic OR “people with aphasia” AND communication OR communicating OR communicate OR conversation OR communication skills AND “life quality” OR “quality of life” OR “well-being” OR well-being OR “life satisfaction” OR QoL. Five databases were selected for the review: Medline Complete, PubMed, PsychINFO, Scopus, and Google Scholar.

This scoping review was conducted based on the Guidelines of Preferred Reporting Items for Systematic Reviews and Meta-Analyses extension for Scoping Review (PRISMA-ScR) to increase methodological transparency [24]. For this study, the PRISMA Checklist was used to endorse better reporting of the methodology and to guide the conduct and reporting of this review. The PRISMA-ScR Checklist aims to support readers in developing a greater understanding of terminology, concepts, and key items to report for scoping reviews [24].

After consensus was reached among the researchers (authors), five inclusion criteria were adopted for this study. Articles were selected for review if the studies: (1) Were published between January 2010 to March 2020 (last 10 years), (2) were written in the English language, (3) included PWA in the chronic stage (6 months post-stroke), (4) reported on assessing quality of life after stroke, and (5) were available with full text access. Predefined study exclusion criteria were (1) studies concerning pediatric stroke, (2) PWA had severe comorbidities, such as dementia and cancer, (3) single case studies, (4) review studies reporting aphasia intervention, and (5) studies reporting on stroke but not on aphasia.

### Extraction of the Data

Two reviewers worked independently to review all articles. Each article was reviewed against the pre-established inclusion and exclusion criteria. Disagreements between the reviewers were resolved by consensus.

## 3. Results

The PRISMA four-stage flow diagram detailing the review process of identification, screening, eligibility, and inclusion of studies [25] was used, as presented in Figure 1. The literature search resulted in 971 articles. Duplicates were discarded to eliminate result bias, which resulted in 952 articles to be screened by title and abstract. Title screening resulted in the removal of 767 additional papers, in which titles did not concern QoL and/or AIR-Qs but instead were related to aphasia rehabilitation. Furthermore, abstract screening resulted in the discarding of a further 140 articles that did not fulfil the inclusion criteria. The final number of studies included in the review were 20 full text articles (see Figure 1). Of these studies, half (10) were related to research on QoL interview-based questionnaires and the other half (10) involved research on AIR-Qs. None of the studies reporting on QoL questionnaires included stroke survivors with or without aphasia in the creation of the questionnaire, and only four of the AIR-Qs studies included PWA in the creation of the questionnaire material. The selected studies shared similar experimental design regarding the qualitative analysis of psychometric assessments, i.e., interview-based questionnaires without any involvement of intervention protocols or single case studies.

Data extracted based on the research questions were: (1) Name of the authors and the year of publication, (2) the country where the research was conducted and the recruitment source, (3) the aim(s), (4) the description of the tools, (5) participant numbers, (6) time post-stroke onset, (7) whether PWA were included in the creation of the tool, (8) whether the tool was piloted on PWA, and (9) the use of related terms of ‘patient involvement’. The data extracted from each of the selected articles are summarized in Table 1.

Studies were evaluated based on whether the term ‘patient involvement’ or variants such as patient partners, peer research, service user, collaborative research, and PPI were used. In addition, studies were examined for pilot testing of the tool on PWA before publication and, if so, at which stage of the process. Selected studies were probed according to their design, methodology, and the involvement of PWA in the creation of these tools (see Table 1). All tools included in the selected studies are considered PRO (patient-reported outcome) or PROMs (patient-reported outcome measures). PRO or PROMs are scales or measurements of the health status of the patients, provided directly by the patient, without the interpretation or involvement of a physician or any other rehabilitation or health specialist [26]. The involvement of PWA in QoL and AIR tools was analyzed and presented as reported in the selected published studies.

### 3.1. Involvement of PWA in the Creation of the QoL and AIR-Q

Of the 20 studies mapped for this scoping review, only four involved PWA during the creation of their tool. Specifically, 10 of the 20 studies (50%) were various adaptations of the Stroke and Aphasia Quality of Life Scale (SAQOL-39) created by Hilary et al. in 2003. These studies included the adaptation of the SAQOL-39 in different languages, as follows: Dutch [27], Greek [28], Portuguese [29], Japanese [30], Singaporean [31], Dutch [32], Chinese [33], Turkish [34], Icelandic [35], and Persian [36]. None of the selected SAQOL adaptation studies involved stroke survivors with or without aphasia in their creation. The original Stroke Specific Quality of Life Scale (SS-QOL) [37] for PWA on which the SAQoL-39 [38] was based, involved extensive consultation with PWA on the content of the pre-established questions, as reported in detail in an earlier study by Hilari and Byng [37], which was not included in our scoping review time window.

The 10 remaining studies related to AIR-Qs research included: (1) The Communication Disability Profile (CDP) [39], (2) the Profile of Life Participation After Stroke and Aphasia (PLALP) [40], (3) the Communication Confidence Rating Scale for Aphasia (CCRSA) (Phase 1) [41], and (4) the CCRSA, Communication Confidence Rating Scale for Aphasia, (Phase 2) [42], (5) the Quality of Life Questionnaire for Aphasia (QLQA) [43], (6) the Assessment for Living with Aphasia (ALA) [44], (7) the Aphasia Communication Outcome Measure (ACOM ) [45], (8) the Measurement of Stroke Environment (MOSE) [8], (9) the Communication Outcome After Stroke Scale for patients and carers (COAST) [46], and (10) the Aphasia Impact Questionnaire (AIQ) [26].

From the 10 selected studies reporting on the development of AIR-Q, only four involved PWA in their creation. These tools were (1) the ALA—Assessment for Living with Aphasia in the study of Simmons-Mackie et al. [44], (2) the ACOM—Aphasia Communication Outcome Measure of Hula et al. [45], (3) the MOSE—Measurement of Stroke Environment in the study of Babulal and Connor [8], and (4) the AIQ—Aphasia Impact Questionnaire of Swinburn et al. [26]. All four AIR-Qs were based on a conceptual framework: The ALA, ACOM, and the MOSE (75%) were constructed on the ICF framework, [6] whereas the AIQ followed the Social Model of Disability [47]. All tools were equated to pre-established gold standard tools (see Table 2).

The PWA that were involved in the four studies as research partners were all adults in the chronic stage post stroke (SPS) (at least 6 months post onset). Sample size ranged from *n* = 329 PWA for the study by Hula et al. [45] on the ACOM, to *n* = 90 PWA for the AIQ [26]. For the MOSES, Babulal and Connor [8] recruited *n* = 43 stroke survivors from which *n* = 21 were PWA, whereas for the ALA of Simmons-Mackie et al. [44], *n* = 101 PWA.

Besides administering standardized psychometric formal assessments, an important evaluation method used across all mapped studies was the patient-reported outcome measures (PROMs) for psychosocial well-being (see Table 3). This would be validated either by the SAQOL [38], for the AIQ, the Porch Index of Communicative Ability PICA [50], for the MOSE the Burden of Stroke Scale (BOSS) [54] and the Visual Analog Self-Esteem Scale (VASES) [55] for the ALA. All assessment procedures used across all studies tapped into the impairment level of the ICF [6] with only two studies, the ACOM and the MOSE, carrying out a functional communication assessment. Both the ACOM [45] and the MOSE [8] performed a functional communication assessment using the ASHA FACS (The American Speech-Language-Hearing Association Functional Assessment of Communication Skills for Adults [56]), which is an observational profile rated exclusively by the clinician [50] and not a PROM.

Methodologies varied among the four studies as participants were assessed on a variety of tools (see Table 3). Two studies, the ALA and the MOSE, used the Western Aphasia Battery (WAB) [57,58] to assess language impairments and only the ACOM study examined the speech mechanism with the Apraxia Battery for Adults-2 [59] and the Dysarthria Examination Battery [60] for PWA who presented additional motor speech disorders. Three studies used the Severity Scale of the Boston Diagnostic Aphasia Examination (BDAE) [49] to define the level of aphasia severity in PWA (For information, Goodglass and Kaplan’s Aphasia Severity Rating Scale; BDAE: Boston Diagnostic Aphasia Examination [49]. Grade 0: “No usable speech or auditory comprehension”. Grade 1: “All communication is through fragmentary expression”. Grade 2: “Conversation on familiar subjects is possible with help from listener”. Grade 3: “The patient can discuss almost all everyday problems with little or no assistance”. Grade 4: “Some obvious loss of fluency in speech or facility of comprehension without significant limitation on idea expressed”. Grade 5: “Minimal discernible speech handicaps”), i.e., for the ALA (MEAN SD: 3.13), for the ACOM (0.72 with severity rating ≥1), and for the AIQ (MEAN SD: 3.35). The MOSE used the BDAE Auditory Comprehension Scale to establish aphasia severity (mean severity score: 68.7). Yet again, besides the MOSE, the other three studies used an emotional state assessment tool, that is, the Geriatric Depression Scale (GDS) [61] for the ACOM and the Communication Associated Psychological Distress Scale of the Burden of Stroke Scale (BOSS CAPD) [54] for the ALA and the AIQ. Severity data suggest that involved PWA could discuss most of the issues related to the questionnaires with little assistance. The authors describe in Table 2 and Table 3 the structure of the QOL and AIR tools as presented in the selected published studies, how they were designed, and how these tools were created in order to demonstrate that PWA were mostly involved as research participants instead of research partners.

### 3.2. The Type of Involvement by PWA When Building the Questionnaires

In this section, an analysis of the type and nature of the involvement of PWA in the creation of the four AIR-Qs will be reported, regarding the nature of the involvement (how PWA were involved), the amount of activity (how often and how much input), and the total contribution of PWA within the selected studies (in which tasks they were involved) (see Table 4).

Based on the PSUE framework [11], for the ALA [44] PWA were involved in the study mostly in the preparatory phase [11]. This means that PWA were recruited in the creation of the ALA in order to report their perspectives on living with aphasia. The preparatory phase included a two-year literature research from the authors and input from the research team and other stakeholders, i.e., speech and language therapists (SLTs) (*n* = 21) and PWA (*n* = 24) and their families, resulting in a 52 self-report questions related to living with aphasia. At this point, PWA were involved in semi-structured interviews as reviewers, in a consultation role. PWA were involved in the selection of the final items to be included in the ALA but they were not involved in the selection of the pictures to support the script. During the execution phase [11], PWA (*n* = 101) were recruited to complete the study. None of the PWA was involved actively in the data collection of the administered interview, but rather they were passive recipients of the ALA administration as study participants. Furthermore, during the execution phase, a focus group with the SLT administrators of the ALA was created to explore experiences during administration of the ALA. A qualitative thematic analysis of emerged themes was performed again, with no involvement of PWA. Finally, PWA were not included in the translational phase [11], which the authors of the study carried out without the involvement of PWA in disseminating, implementing, or evaluating the study outcomes.

In the ACOM [45], PWA were not included in the preparatory phase [11]. The authors performed a review of existing instruments. After the literature review, 426 items were pooled and selected to be judged by three SLTs. The resulting 211 items were then grouped based on the seven domains of the ICF framework [6], again without any involvement of PWA. In the *execution phase* [11], these items were surveyed separately in small groups (3–5 people) of PWA (*n* = 59) and their communication partners (*n* = 61) to rate each item as ‘very’ or ‘not at all important’ to their lives, in a consultation role. Similar data were collected from SLTs (*n* = 114) via a Web-based survey. When all data were selected, researchers modified 11 items, eliminated 52 items, and added 7 items judged by stakeholders (PWA and surrogates) as unambiguous or relevant to their lives. Neither PWA nor other laypersons related to them were involved in the analysis of the results nor during the translational phase [11], which was carried out exclusively by the authors (dissemination, implementation, and evaluation) of the study.

In the MOSE study [8], PWA were involved straight to the execution phase [11] of the study, which was named *phase one* by the authors. Phase one was completed in three steps. The first step included an interview and an initial development of the tool with input from key informants, that is, PWA (*n* = 5) of mild severity. The interview sessions were carried out using an aphasia-friendly format and, after being videotaped, were transcribed and coded by two individual coders, respectively. Thematic analysis was then performed based on Grounded Theory by SLTs without the involvement of PWA or surrogates [8]. Themes emerged when the concepts were discussed with all participants. During the second phase of the MOSE study, PWA were neither involved in data analysis and interpretation of findings nor in the third phase of the research process, the translational phase [11], which was again carried out by the authors and the research team.

For the AIQ [26], PWA were involved in the preparatory and execution phases [11] of the research through co-production methodology. The AIQ prototype was developed with the Disability Questionnaire (DQ) within the Comprehensive Aphasia Test (CAT) [63] but without the direct involvement of PWA. After a couple of years, the authors formed an *advisory group* including PWA to revise the DQ in the Communication Disability Profile (CPD) [64] and to reflect on the content. In co-production with PWA, the CPD was then created, which is used as a PROM to measure QoL of PWA in clinical practice and research. In 2011, the CDP underwent a revision from PWA and the research team, which resulted in the AIQ prototype. For the preparatory phase [11] of the AIQ, PWA (*n* = 6) were recruited to form the AIQ Development Group. This group revised the 56 CPD items and advised on format, content, and scoring. Then the authors field-tested the selected items with PWA (number is not clearly defined by the authors). PWA chose all AIQ items, and SLT administrators gave feedback on the items selected by the AIQ Development Group. In the execution phase [11], PWA (*n* = 31) were recruited to participate in *stage one* for testing the AIQ (i.e., AIQ prototype validation). PWA did not participate in data analysis or in the interpretation of results of this stage. In the second stage (i.e., the AIQ-21 validation), the same group of PWA were recruited for both tests: Concurrent validity and internal consistency. Twenty PWA were recruited as participants to complete the psychometric testing of the AIQ-21. None of the PWA in this stage had been involved in the AIQ prototype testing as research partners [26]. Once more, PWA did not participate in data analysis and interpretation of results or the *translational phase* [11].

A summary of the involvement of PWA in the four selected studies, based on the PSUE framework [11], is shown in Table 5.

### 3.3. Tools Pilot Tested on PWA before Publication

From the 10 selected SAQOL adaptation studies, only two (20%) used a pilot study while adapting the SAQOL to their target population. The study by Qiu et al. [33] for the Chinese adaptation of the SAQOL partially piloted the tool mainly for the cultural adaptation of the questionnaire. This consisted of two phases. Phase one involved two stroke survivors with no aphasia, two PWA post-stroke, and two caregivers. The second phase included PWA (*n* = 5) and stroke survivors (*n* = 5) with no aphasia. Second, in the study by van Ewijk et al. [32] the Dutch adaptation of the English SAQOL [38] was piloted in three stages: A pretest stage that included PWA (*n* = 13), a second phase with PWA (*n* = 47), and a test-retest reliability method which included PWA (*n* = 35). From the literature on the AIR-Qs, only four of the 10 (40%) selected studies had piloted the tools prior to publication. Again, these tools were the ALA, the ACOM, the MOSE, and the AIQ. The ALA [44] was piloted six times in a period of 15 months with a total number of 48 PWA and 5 SLTs. The ACOM [45] was pilot tested with two different groups: *n*1 = 59 PWA and *n*2 = 61 communication partners. For the MOSE [8], the authors underwent a three-stage pilot study: First pilot with 10 PWA diagnosed with mild to moderate aphasia, second pilot with 10 PWA and 10 stroke survivors without aphasia, and the third pilot stage included 10 PWA and 10 stroke survivors without aphasia. The AIQ [26] was piloted on 31 PWA in total, before publication.

### 3.4. The Occurrence of the Term ‘Involvement’ in the Selected Studies

The terminology used to describe ‘patient involvement’ in this review varied across the studies. The study by Simmons-Mackie et al. [44] reported PWA as ‘stakeholders’ in the ALA. Swinburn et al. [26] used the terms ‘user involvement’, ‘in partnership with PWA’ and ‘research partners’ for the AIQ. In the MOSE study, Babulal and Connor [8] used the phrase ‘PWA participating in research’ to describe their involvement in the study without providing a definition or further explanation. Finally, the study by Hula et al. [45] used the term ‘Patient Centered Outcome Research’ for the ACOM. There was no similarity in terms or consistency in the use of a specific term or a definition to describe the involvement of PWA between the selected studies. In fact, terminology in the current literature is imprecise and variable for describing the involvement of PWA in research.

### 3.5. Determination of Any Additional Issues That PWA Would Bring to the Creation of QoL and/or AIR-Qs

In the attempt to extract and present the purpose and outcomes of the involvement of PWA in the selected published studies, we endeavored to identify the positive impact of research-oriented PPI in the stroke-related aphasia studies. However, in many cases, this was not feasible as this information was missing from the selected published studies. As a result, it is fair to suggest that PPI has not been fostered in the development of aphasia research protocols during the last decade. The impact of PPI in aphasia research might include the effects on the patients themselves, any other layperson (family members, carers, advocates, patients’ organizations), and the community in general. The range of possible impact results is included in the GRIPP2 [64], which aims to improve the reporting of PPI and the grounds for discussing and evaluating the impact of PPI in general.

None of the selected studies of this review involved PWA and other laypersons consistently throughout the research phases and stages. In all mapped studies, PWA were mainly involved in semi-structured interviews in a consultation role, in which they revised or discussed pre-established questionnaires about QoL after stroke or the impact of aphasia on their everyday life. There is a gap in the aphasia literature concerning a specific framework that could elucidate important data from the involvement of PWA in research, i.e., the type of involvement, the amount of involvement, and the contribution of PWA, and all these factors affect the research outcome. Obviously, there is a lack of consistency in adhering to a functional framework in qualitative participatory aphasia research. Potentially, the regular involvement of PWA in research studies could cover a number of additional issues in qualitative health-related participatory research, such as:(1)Tailored research priorities/questions: The consistent involvement of PWA in research will foster the setting of tailored research priorities, themes, and subjects and improve research conduct [65]. Conducting a study on a special interest topic or a common area of concern revealed by PWA will promote realistic research questions, functional outcomes, and promote better living with aphasia. The aim of consistent involvement of PWA in the creation of such tools is to co-produce meaningful knowledge and methodological consistency and to give a voice to PWA who are often excluded from research, especially on topics like QoL, which are very subjective and sensitive to measure. PWA and other laypersons can identify the gaps and help to formulate targeted research questions. The involvement of PWA will also optimize the validity, design, and applicability of the research itself and the effectiveness of the resulting tools [11]. Co-production of evidence that is both scientifically robust and patient-specific oriented is important to appraising healthcare professionals’ and rehabilitation specialists’ practice [66].(2)Equalizing the power position-shift of power: The consistent involvement of PWA in research groups will promote a balance in power relations, between experts and PWA. With the re-establishment of the patient’s position, from passive receiver of pre-established scientific input/data to the position of the “expert” directly living with condition, researchers create an atmosphere of acceptance [67]. This approach will reveal different layers of understanding of aphasia as a symptom and the aphasic as the person carrying the symptom. Regular involvement of PWA promotes patient centeredness and a focus on specific concerns [67]. Involvement of PWA in qualitative participatory health research will promote a more moral/ethical way to empower PWA in an otherwise expert-dominated endeavor [11].(3)Endorse result presentations, promote dissemination, and ensure research impact: The consistent involvement of PWA promotes strategies for preventing and handling missing data [66,68]. PWA can contribute to creating reports, outcomes, and research results in a more comprehensible and aphasia-friendly format. This practice ensures data integrity and rigorous analyses. Dissemination of the research results between the stroke community, all involved agencies (stroke support organizations, aphasia associations, patients’ advocates, politicians), and the healthcare rehabilitation specialists’ ecosystem will be easily manifested [69]. The exploration of a research topic of mutual interest to both expert scientists and PWA will strengthen the impact of the research [65,67] in both the scientific society and the community and avoid research waste by funding agencies [68].(4)Best practice in aphasia research and rehabilitation: The consistent involvement of PWA will enhance quality, relevance, and acceptability to all involved stakeholders [18,64]. This will lead to the development of an evidence base in the field of stroke aphasia rehabilitation that will facilitate more effective synthesis of research protocols in the future. This practice is essential in transforming the healthcare system to be more patient-centered and sustainable [17].(5)Meeting funders’ demands: The new trend in healthcare research is the obligatory demonstration of the direct involvement of patients and other laypersons in research proposals to ensure civically responsible and moral research [11]. Along with the involvement of PWA, the inclusion of national aphasia associations and stroke support organizations for the dissemination and promoting of the research results ensures the sustainability of the projects. Involvement of PWA in research protocols should be an obligatory requirement in contemporary healthcare research proposals [69].

## 4. Discussion

The current scoping review synthesizes the involvement of PWA in the creation of QoL and AIR-Qs, as reported in the selected published studies. The aim was to determine the evidence showing the direct involvement of PWA in the creation of QoL and AIR-Qs. The consistent involvement of PWA as research partners in aphasia research is a novel concept, as demonstrated by the results, i.e., PWA were mainly excluded from research teams in 75% of the studies included in this scoping review or were partly involved in collaborative research relationships.

PWA were partially or loosely involved in the selected published studies. In the ALA study [44], PWA were mostly involved in taking part in semi-structured interviews as members of a working group and having a consultation role within the research team. None of the PWA were involved in data collection, data interpretation, or in the management phases of the research (i.e., dissemination, implementation, sustainability). Similarly, in the ACOM study [45], PWA had a simple advisory role in the first phase of the study. In the same way, for the MOSE study [8], PWA were involved in the first stage of the study as key informants to revise selected material. Finally, in the AIQ study [26], PWA were involved in more phases [11] of the research through co-production methodology. PWA served as an advisory group to revise questionnaires, decide on which items to be included in the tool, and advise on format, content, and scoring procedures. Nevertheless, PWA did not participate in setting the research questions, did not propose the themes to be involved in, and were not included in analyzing data or in the interpretation of the results. The main limitations of these studies are their mixed methodology of PPI for PWA and their noncomparative nature. There is a lack of a standardized procedure in reporting PPI processes in research involving PWA as patient stakeholders, which is an additional limitation to extrapolating the evidence. Additionally, no representatives of PWA or other laypersons were actually co-authors on the study manuscripts or were involved in the preparation, design, analysis, or dissemination of the selected studies.

The engagement of PWA in the selected studies is problematic and shows that researchers involve PWA mainly as consultants or advisors, after researchers have created the content of the questionnaires. The reason for placing PWA in the position of the ‘advisor’ and/or the ‘consultant’ results from the fact that researchers do not have the resources, methodologies, and frameworks on how to train, work, and collaborate with people with acquired communication impairments [70]. There are misconceptions in the scientific society on PWA’s mental capacity and their ability to give informed consent, as their spoken and written language abilities are affected by stroke. Brady et al. [70] stated that there are three different groups of PWA when examining their capacity to give informed consent and make decisions. On the one extreme, there is a group of people with very mild aphasia who have full capacity to make their own decisions and participate in research projects with informed consent. On the other end, there is another group of people with very severe aphasia (alongside significant cognitive impairments) whose capacity to make informed decisions has been severely affected and are fully dependent on authorized guardians [71]. Somewhere in the middle, there is a third group of PWA who have retained the capacity to make informed decisions, but this capacity is obscured by their language difficulties [71]. There is no doubt that research studies involve a large amount of verbal and written material to manage and analyse, and this is the most challenging area for the involvement of PWA in the research process. Any group of PWA can and should be included in research projects to assist with the generation of meaningful evidence with ‘reasonable adjustments’ (i.e., text modification, peer support groups, communication support partners) and other evidence-based strategies that can maximize communication [70,71]. In the position paper by Stein and Wagner (2006) it is recommended to use a ‘facilitated consent’ model when including PWA in the consent process. This proposed model suggests that PWA could be actively involved in informed consent procedures using a proxy that they designate and will be ‘used’ as a helper, an advisor, and a communication facilitator when making decisions or discussing specific topics with medics [72]. This model could also be applied in qualitative research studies, in which PWA assign their communication helpers throughout the research process.

How aphasia and stroke impact the lives of PWA has been studied for many years, and now it is time for PWA themselves to set their own research priorities and explore specific issues that usually do not attract public funding. Research proposals formulated by PWA will be of high-impact value not only in the society of the rehabilitation experts but especially to the stroke survivors’ community. Positive effects of studies that consistently include PWA will provide confidence and expectations for PWA to express their needs and get tailored rehabilitation. Personalized and qualitative therapeutic goals will enable rehabilitation specialists to gain more insight into the communication barriers chronic stroke survivors with aphasia face, which reduce their social integration. The American Speech and Hearing Association—Quality of Communication Life Scale [73] suggests that “the more positive the personal and environmental factors, the more successful the [person’s] communication acts, the better the quality of communication life” (p. 2).

Taking into consideration the perspectives of PWA, as service users, in all phases of the research, is critical to generating findings that will accelerate translation to real-world clinical practice and promote functional interventions and strategies for living successfully with aphasia (activity and participation level: ICF, 2001). According to the Five Good Communication Standards of the Royal College of Speech and Language Therapists [74], the individual risk of having a communication difficulty means PWA are misunderstood and experience failure and exclusion from events, activities, and relationships. Good communication only exists as part of positive everyday relationships, boosting self-esteem and success. Good communication crosses all dimensions of care, support, and enablement. Without good communication PWA struggle to learn, achieve, and make friends, all fundamental for citizenship and central to improving quality of life. Communication quality in PWA is defined based on the following: (1) Involvement with decisions about their care, (2) making choices about daily life activities, (3) creating opportunities to communicate needs and thoughts, (4) to be understood and able to express their wants in relation to their health and well-being, and (5) being treated with respect and dignity [74]. PWA should have the opportunity to establish good communication with the scientific society as well.

### 4.1. Strengths and Limitations of This Review

The strength of this review is the use of recommended and rigorous methodology widely accepted in the conduct of scoping reviews. We used broad search terms across a range of databases in order to exploit the possibility of including all the available research involving PWA in the creation of QoL and AIR-Qs. Nevertheless, the variability in the term ‘patient involvement’ in PWA and the lack of a definition may have restricted the search process as important concepts might have been ignored when determining the search terms. Another limitation is the relatively small number of studies that reveal the involvement of PWA in research, which makes it difficult to profile the engagement of PWA in research studies. Also, the fact that the included studies involved PWA, mainly in the primary stages of the research, focused mainly in using PWA as ‘counselors’ and ‘advisors’, restricting the in-depth analysis of the review outcomes. This study was based on scoping review methods and not a systematic review. A search spanning the last decade was performed, which is the common process used in scoping review methodology in health sciences. The researchers (authors) of this study explored the latest information, the trends, and the knowledge gaps on the proposed topic in current bibliography and not in bibliography in general. A systematic review could possibly have a broader time-period search and include additional published studies with valuable information, but this was not the purpose of this scoping review.

### 4.2. Knowledge Gaps and Future Recommendations

A major challenge in PPI of PWA is the ‘approach’ of how to put it into practice. There is an absence of standard approaches and frameworks conducting and reporting PPI with PWA, which limits the potential for indexing, knowledge synthesis, and comparative effectiveness to determine best practices. Additionally, this review revealed an inconsistency in the term used by the published studies in involving PWA as research partners. A future recommendation is that researchers who are translating, adapting, and validating pre-established QOL and AIR tools into other languages, involve PWA and their communication partners in their study protocols and as research partners throughout the study using the GRIPP2 reporting checklist [64] or any other relevant framework. Future research should focus on the creation of a comprehensive conceptual framework for qualitative participatory approach in aphasia research, which is meaningful to PWA and engages them in research partnership within each research phase. There is a strong need for the creation of such a functional methodological framework based on foundational engagement principles to facilitate patient-centered qualitative research design for people with communication impairments.

## Figures and Tables

**Figure 1 brainsci-10-00688-f001:**
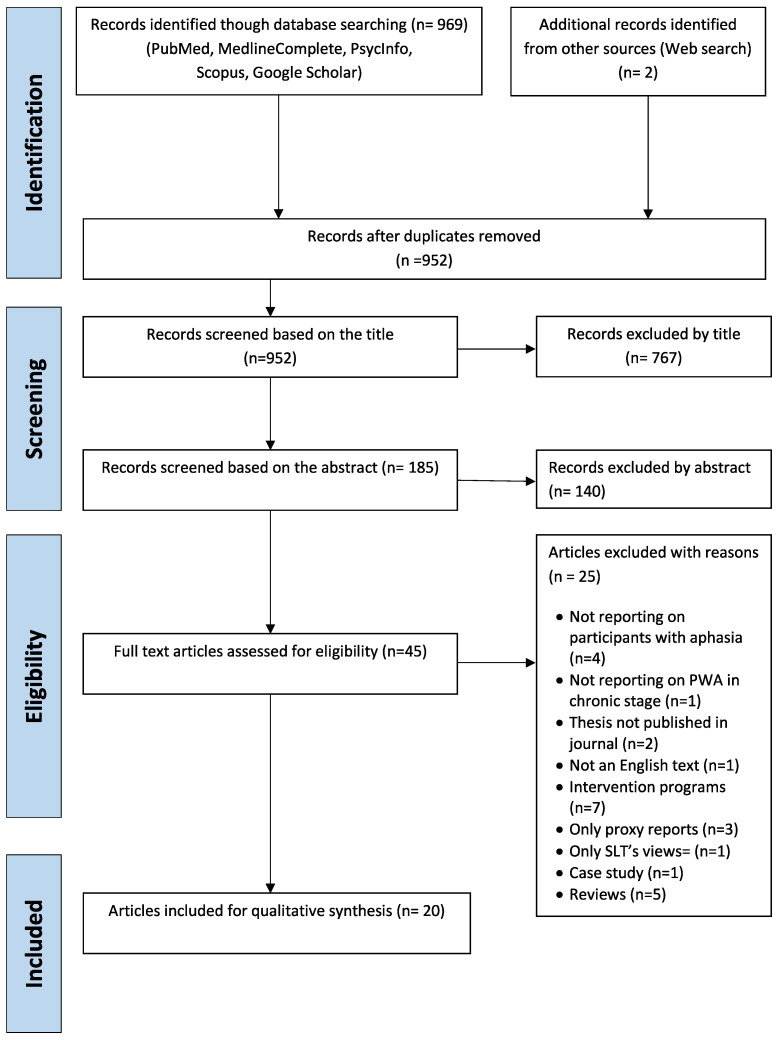
PRISMA flow diagram for the scoping review process.

**Table 1 brainsci-10-00688-t001:** Summary table of quality-of-life and aphasia impact-related studies for people with aphasia (PWA) in chronological order.

#	Author Year of Publication	Country Recruitment Source	Aim (s)	Tool Name and Description	Number of Participants	Time Post Stroke Onset	PWA Included in the Creation of the Tool	Pilot Tested with PWA	Terms Related to Patient Involvement
1	Chue et al., 2010	Australia: The Australian Aphasia Association and the Stroke Association Victoria	To investigate the test–retest reliability and internal consistency of the Activities, Participation, and Emotions sections of the Communication Disability Profile (CDP)	The CDP is an outcome measure that includes aphasia-friendly design features (e.g., pictures, simple wording, key words in bold, picture-rating scales) to support PWA in self-reporting the impact of aphasia on their lives.	*n* = 16 PWA	Chronic Stage	No	No	No related terms
2	Manders et al., 2010	Belgium: Rehabilitation Centers and Hospitals in Belgium	To examine the quality of life (QoL) of people with aphasia and to the influence of variables such as age, time post onset and (degree of) social support on the QoL of aphasic persons.	Stroke and Aphasia Quality of Life Scale (SAQOL-39) is an interview-based psychometric tool for stroke survivors with or without aphasia	*n* = 129 *n*1 = 43 PWA *n*2 = 43 people with Acquired Brain Injury, no aphasia *n*3 = 43 healthy controls	Subacute and Chronic Stage	No	No	No related terms
3	Wallace, 2010	United Sates: Referrals from SLT’s, physical therapists, physicians, stroke support groups in Ohio	To obtain authentic information about life participation after stroke and aphasia.	Profile of Life Participation After Stroke and Aphasia (PLALP) is a semi structured, conversational approach to obtain self-reported information about a person’s life participation profile	*n* = 40 PWA	Chronic Stage	No	No	No related terms
4	Cherney et al., 2011	United States: Center for Aphasia Research and Treatment at the Rehabilitation Institute of Chicago	To describe the first phase in the development of the CCRSA.	Communication Confidence Rating Scale for Aphasia (CCRSA)—self rating questionnaire with 10 item visual analog scale	*n* = 21 PWA Chronic Stage	Chronic Stage	No	No	No related terms
5	Babbitt et al., 2011	United States: Variety of settings in Chicago	To report data from the second phase of the project in which the CCRSA was revised to include 10 items.	CCRSA was developed by asking PWA to self-rate their communication confidence.	*n* = 94 PWA	Chronic Stage	No	No	No related terms
6	Efstratiadou et al., 2012	Greece and Cyprus: SLTs and neurologists working for the national health system or in private practice in Greece and Cyprus	To explore the acceptability, test-retest reliability, internal consistency and construct validity of the Greek SAQOL-39g in a stroke population, comprising people with and without aphasia	Stroke and Aphasia Quality of Life Scale (SAQOL-39g) Greek version is an interview-based psychometric tool for stroke survivors with or without aphasia	*n* = 86 with stroke *n*1 = 62 stroke survivors without aphasia *n*2 = 24 PWA	Chronic Stage	No	No	No related terms
7	Rodrigues Leal, 2013	Portugal: Four speech and language therapy centers	To translate and assess the psychometric properties and reliability of the Portuguese version of the SAQOL-39 in a group of chronic aphasia patients	Stroke and Aphasia Quality of Life Scale (SAQOL-39) is an interview-based psychometric tool for stroke survivors with or without aphasia	*n* = 33 PWA	Chronic Stage	No	No	No related terms
8	Spaccavento et al., 2013	Italy: Italian Aphasia Association in Puglia	To draw up a Quality of Life questionnaire for aphasics (QLQA) focusing particularly on difficulties in interpersonal relationships and on the loss of independence because of language disorders	QLQA I an interview-based psychometric tool.	*n* = 183 *n*1 = 146 PWA *n*2 = 37 controls	Chronic Stage	No	No	No related terms
9	Simmons-Mackie et al., 2014	Canada: Outpatient services of the Aphasia Institute in Toronto	To assess test-retest reliability, construct validity of the Assessment for Living with Aphasia (ALA) and the ability to discriminate aphasia severity.	ALA is a patient-reported aphasia-friendly pictographic measure assessing aphasia, participation in life situations, environment facilitators and barriers to communication, personal factors, and overall QoL in an interview format appropriate for use with severe aphasia.	*n* = 101 PWA	Chronic Stage	Yes Via Focus groups *n*1 = 24 PWA *n*2 = 21SLTs	Yes 6 pilots (15-month period) *n*1 = 48 PWA *n*2 = 5 SLTs	Stakeholders
10	Hula et al., 2015	United States: The greater metropolitan areas of Minneapolis, Pittsburgh etc., from clinics and hospitals and local stroke support groups, the Healthcare System Audiology and Speech Pathology Research Registry and the Western Pennsylvania Participant Registry of University of Pittsburgh	To investigate the structure and measurement properties of the Aphasia Communication Outcome Measure (ACOM)	The ACOM is a patient reported outcome measure of communicative functioning for persons with aphasia.	*n*1 = 329 PWA *n*2 = 329 associated surrogates	Chronic Stage	YES	YES *n*1 = 59 PWA *n*2 = 61 communication partners	Patient-Centered Outcomes Research
11	Kamiya et al., 2015	Japan: 4 settings: 3 speech and language therapy services and 1 non-profit organization for people with aphasia	To validate the Japanese version of SAQOL-39, designated as SAQOL-39-J, and compare the scores among different types of aphasia	Stroke and Aphasia Quality of Life Scale (SAQOL-39-J) Japanese version, is an interview-based psychometric tool for stroke survivors with or without aphasia	*n* = 54 PWA	Chronic Stage	No	No	No related terms
12	Bambini et al., 2016	Italy: Outpatient services in ambulatory settings and inpatients in the Neurorehabilitation Unit in Pavia	To validate the COAST and Carer COAST scales for the Italian-speaking population; to explore the applicability of the COAST scales to a wider range of people with communication problems not limited to moderate aphasia; to explore the agreement between patient’s and carer’s perspective on communication difficulties, and the effect of severity	The Communication Outcome After Stroke Scale for patients and carers (COAST and Carer (COAST) are scales that are comprised of two components, interactive communication skills and their impact on quality of life, assessed through 20 question items, from the point of view of patient and carer.	*n*1 = 30 PWA *n*2 = 28 carers	Chronic Stage	No	No	No related terms
13	Babulal and Connor, 2016	United States: Stroke Registry of the Cognitive Rehabilitation Research Group at Washington University School of Medicine	To present the development and psychometric properties of a new environmental measure that identifies barriers and facilitators in receptivity, physical environment and communication for post-stroke populations including survivors with aphasia.	The Measure Of Stroke Environment (MOSE): stroke-specific measure of the environment, in an aphasia friendly format, evaluating under-assessed aspects of the environment that contribute to participation limitations in post-stroke survivors.	*n* = 43 stroke survivors *n*1 = 24 PWA *n*2 = 19 stroke survivors without aphasia	Chronic Stage	YES *n* = 5 PWA	Yes 1st pilot *n* = 10 PWA 2nd pilot *n* = 10 PWA *n* = 10 stroke survivors 3rd pilot *n* = 10 PWA *n* = 10 stroke survivors	PWA participatingin research
14	Calis et al., 2016	Turkey: Neurology department of governmental hospital	To translate the SAQOL-39 into the Turkish language (SAQOL-39/TR) and assess its reliability and validity in patients who had aphasia	Stroke and Aphasia Quality of Life Scale (SAQOL-39/TR) Turkish version, is an interview-based psychometric tool for stroke survivors with or without aphasia	*n* = 40 PWA *n* = 22 controls with dysarthria	Chronic Stage	No	No	No related terms
15	Guo et al., 2016	Singapore: Community	To compare outcomes between stroke survivors with and without aphasia in Singapore and examine the sensitivity and responsiveness to change of the Stroke and Aphasia QOL Scale (SAQOL-39g) and its Singapore (Mandarin) variant, SAQOL-CSg	Stroke and Aphasia Quality of Life Scale (SAQOL-39-CSg) Singapore (Mandarin) version, is an interview-based psychometric tool for stroke survivors with or without aphasia	*n* = 94 Stroke survivors *n* = 65 no aphasia *n* = 29 PWA And *n* = 78 Stroke survivors *n* = 55 no aphasia *n* = 23 PWA	Subacute phase (3 months) chronic stage (12 months)	No	No	No related terms
16	van Ewijk et al., 2016	Netherlands: Six aphasia centers (Almere/Bussum, Drachten, Leeuwarden, Terneuzen, Tilburg and Utrecht).	To adapt the English Stroke and Aphasia Quality of Life—39 item generic stroke scale (SAQOL-39g) into Dutch. To investigate the psychometric properties (acceptability, internal consistency, test–retest reliability and construct validity) of the Dutch version (SAQOL-39NL)	Stroke and Aphasia Quality of Life Scale (SAQOL-39NL) Dutch version, is an interview-based psychometric tool for stroke survivors with or without aphasia	*n* = 60 PWA	Chronic Stage	No	Yes Pre-test *n* = 13 PWA Phase II *n* = 47 PWA Test–retest *n* = 35 PWA	No related terms
17	Swinburn et al., 2018	United Kingdom: Connect Center London	To report the quantitative aspects of a mixed methods study that developed and validated a concise PROM, the Aphasia Impact Questionnaire (AIQ), co-produced with People with Aphasia (PWA)	The AIQ is a subjective, pictorial, self-report questionnaire. It is divided into 3 sections, each containing questions exploring domains of living with aphasia: communication; participation; and well-being/emotional state.	*n* = 90 PWA	Chronic Stage	Yes	Yes *n* = 31 PWA	‘In partnership with PWA’ ‘User Involvement’ ‘PWA Research Partners’
18	Qiu et al., 2019	China: Rehabilitation Medicine Department of the Affiliated Hospital of Sun Yat-sen University and Panyu Central Hospital	To develop a Chinese-version of the Stroke and Aphasia Quality of Life-39 generic version (SAQOL-39g) and evaluate its feasibility, reliability, and validity in Chinese patients with stroke-induced aphasia	SAQOL-39g is an interview-based psychometric tool for stroke survivors with or without aphasia	*n* = 84 PWA *n* = 60 PWA mild/moderate *n* = 24 PWA severe aphasia *n* = 82 their proxies	Chronic Stage	No	Yes Phase one: *n* = 2 stroke survivors no aphasia *n* = 2 PWA after stroke *n* = 2 caregivers Phase two: *n* = 5 PWA *n* = 5 stroke survivors no aphasia	No related terms
19	Kristinsson and Halldorsdottir, 2020	Iceland: Local Stroke Support Organization	To translate and adapt the SAQOL-39g into Icelandic and examine its psychometric properties. To gather preliminary information on the health-related quality of life of stroke patients in Iceland	Stroke and Aphasia Quality of Life Scale (SAQOL-39g) is an interview-based psychometric tool for stroke survivors with or without aphasia	*n* = 20 stroke survivors *n*1 = 10 stroke survivors without aphasia *n* = 10 PWA	Chronic Stage	No	No	No related terms
20	Azizbeigi-Boukani et al., 2020	Iran: Shariati Hospital, a referral center for stroke in Tehran and private Clinics	The aim of this study was to examine the reliability and validity of the Persian version of the SAQOL-39, and to examine the agreement between the self- and proxy-report versions of the scale	Stroke and Aphasia Quality of Life Scale (SAQOL-39) is an interview-based psychometric tool for stroke survivors with or without aphasia	*n* = 20 stroke survivors	Chronic Stage (*n* = 20) and Acute stage (*n* = 10)	No	No	No related terms

**Table 2 brainsci-10-00688-t002:** Summary of the conceptual frameworks and gold standard tools of the four studies that included PWA.

#	Tool	Conceptual Framework	Gold Standard Measure
1	ALA [44]	A-FROM [48] and the ICF [6] ALA was designed into a priori 4 domains of the ICF (1) Language impairment (2) participation (3) environment (4) personal	A-FROM [48]
2	ACOM [45]	ACOM was based on “Functional Communication”, a concept which includes a person’s ability to effectively convey and receive personally relevant messages regardless of modality and to do so in his or her natural environment ACOM was analyzed based on the seven domains of the ICF [6] (1) Community Life (2) Domestic Life (3) Economic Life (4) Education and Work (5) Interpersonal and Leisure (6) Recreational and (7) Self-Care	-Boston Diagnostic Aphasia Examination (BDAE) Severity Scale [49] -PICA (Porch Index of Communicative Ability) [50] -ASHA FACS (The American Speech-Language-Hearing Association Functional Assessment of Communication Skills for Adults; [51])
3	MOSE [8]	ICF based [6]: was designed into a priori domains of the ICF framework	-The Stroke Impact Scale (SIS) version 4.0 [52]—National Institute of Health Stroke Scale (NIHSS) [53]
4	AIQ [26]	Social Model of Disability by Byng and Duchan [47]	Burden of Stroke Scale (BOSS) [54]

**Table 3 brainsci-10-00688-t003:** Assessment tools used in the four Aphasia Impact Related studies selected in the review.

#	AIR-Q	Language Assessment Tools	Speech Assessment Tools	Aphasia Severity Definition	Psychosocial Well-Being Assessment	Patient Reported Outcome Measures (PROMs)	Functional Communication Assessment
1	ALA [44]	WAB [56]	None	Severity Scale of BDAE [49]	BOSS CAPD [55]	VASES [58] SAQOL [38]	None
2	ACOM [45]	Arizona Battery for Communication Disorders [62] WAB Revised [57]	Apraxia Battery for Adults-2 [59] Dysarthria Examination Battery [60]	Severity Scale BDAE [49]	GDS [61]	BOSS [53] PICA [50]	ASHA FACS [51]
3	MOSE [8]	None	None	BDAE Auditory Compreh. Scale [49]	None	PICA [50]	ASHA FACS [51]
4	AIQ [26]	None	None	Severity Scale BDAE [49]	BOSS CAPD [55]	BOSS [53]	None

**Table 4 brainsci-10-00688-t004:** The type of involvement of PWA in the creation of the four selected studies.

#	Tool	Nature of Involvement	Amount of Activity	Contribution of PWA
1	ALA [44]	Consultation Role	Semi-structured interviews	Item analysis and selection for ALA
2	ACOM [45]	Consultants PWA and communication partners	Small groups surveys	Item selection for ACOM
3	MOSE [8]	Key informants	Interview sessions	Provided information and experiences during interview sessions
4	AIQ [26]	Advisory group, AIQ Development Group, AIQ field testing group, Statistical testing group, item selection panel	One off group (1), one off interviews (1), group meetings (4), one off assessments (2)	Selection of all 21 AIQ items advise on format, content, scoring data provision as research participant

**Table 5 brainsci-10-00688-t005:** Summary of the phases and stages of the involvement of PWA based on the Patient and Service User Engagement (PSUE) framework [11].

#	Name of Tool and Authors	Preparatory Phase	Execution Phase	Translational Phase
1	ALA [44]	YES	NO	NO
2	ACOM [45]	NO	YES partially	NO
3	MOSE [8]	NO	YES partially PWA as key informants	NO
4	AIQ [26]	YES partially AIQ Development Group	YES partially AIQ Development Group	NO

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
