# Peer review of "Are People with Aphasia (PWA) Involved in the Creation of Quality of Life and Aphasia Impact-Related Questionnaires? A Scoping Review"

_brainsci, 2020, doi:10.3390/brainsci10100688_

Round 1
Reviewer 1 Report
Thank you for the opportunity to review this interesting paper. It is on two topics that closely align with my interests - quality of life and involvement of users. The premise of the paper is an important and significant one, and as such, this paper would be welcomed in the research community. The authors have drawn in acceptable scoping review frameworks, reporting (PRISMA-ScopingReview) and a 3 phase framework for analysing involvement which is good, and the style of writing is clear and accessible.
However, there is one fundamental issue which needs to be recognised, and I have a further strong concern, that affect the paper as it stands currently; and then there are additional ways in which the paper could be strengthened. Firstly, the authors need to acknowledge that using research articles as a means for assessing involvement is reliant on the reporting of that involvement in the said articles – so it is not possible on this basis to claim that various quality of life tools or aphasia-impact-related tools have not involved people with aphasia in their development. This is because reporting (of many and varied aspects of the research process) is often incomplete or entirely omitted at the publication stage; this does not mean it did not happen. For example, in this paper, the authors state that the original development of the SAQOL-39 did not involve stroke survivors with aphasia based on their reading of reference #37 (2003). I am not the author of this QOL tool, but I know it well and I know this claim to be inaccurate regarding the tool. An earlier paper (2001) https://pubmed.ncbi.nlm.nih.gov/11340850/ highlights two different ways people with aphasia were involved, and indeed, the SSQOL on which the SAQOL was developed initially did involve stroke survivors which was one of the main reasons that Hilari selected the SSQOL for adaptation as her PhD. As such, it is essential that the authors more carefully explain their data is derived from reference to specific research articles, rather than making claims about the tool itself. Authors need to very clearly acknowledge this throughout the entire paper and soften their statements in recognition of this. So, the paper needs to orient towards this as assessing the involvement of PWA in instrument development as reported in published studies. Secondly, the authors need to provide a credible rationale for why the January 2010 start date for their search; there is no viable reason that exists in the field, and indeed QOL tools have been in development since 2000. Had their search started earlier, they would have found the 2001 publication above in their search. Finally, the authors are advised to consider content validity in relation to the premise of their paper. QOL and other tools are developed in accordance with various psychometric testing and properties, of which content validity is relevant to the topic of this paper. This reference for example highlights how it could strengthen the argument the authors are making: https://www.cosmin.nl/wp-content/uploads/COSMIN-methodology-for-content-validity-user-manual-v1.pdf
Additionally, authors are advised to consider the reporting standards now available for PPI https://www.equator-network.org/reporting-guidelines/gripp2-reporting-checklists-tools-to-improve-reporting-of-patient-and-public-involvement-in-research/ Addressing this fundamental issue, concern, and these two additional points would really strengthen this paper for publication and make it considerably more impactful in the research community. I hope the authors can see the value and importance of this and I would be happy to see a revision of this manuscript.
Other Genera Comments
The introduction takes a strong exclusion stance, which is needed to create the argument for this research. However, it does not reflect at least a decade of research in aphasia rehabilitation research where researchers have focused almost entirely on clients’ or users’ views (e.g. Kyla Brown’s research on living successfully with aphasia; Sarah Wallace’s research on prioritised outcomes of rehabilitation research; Molly Manning’s very recent review). It is slightly disingenuous not to consider this. The need though to investigate user contribution to QOL and aphasia impact related questionnaires is a valid one.
Reliable methods have been used, and reporting guidelines used to direct the research.
It is unclear why authors have chosen a January 2010 start date for their search – no rationale is provided and there is none that seems obvious. They needed to run searches in the aphasia research literature to appropriately identify when QOL and impact questionnaires started being used, and subsequently set a much earlier date as start date OR impose no start date (which would be more interesting and more defensible). Indeed Quinting et al 2018 might have been at minimum useful in exploring date criteria as a short-cut to identifying QOL tools: https://www.tandfonline.com/doi/abs/10.1080/02687038.2018.1486388
In the results, there’s no real need to use both abbreviations and full test names repeatedly. Abbreviations would suffice after first fully spelled instance of the QOL or AIR tool. In Table 2, please consider adding a line space between rows – it is currently difficult to see where one description of a tool’s conceptual framework begins and ends; the same applies for Table 1. The results report more data than is described as being extracted in lines 164-169. Research question 3 is also missing. So Table 2 reports on gold standard measures which relates to psychometrics comparisons that the papers would have reported on if psychometrics were a focus of the paper, so it’s unclear why this relates to PPI reporting in this study. Similarly in the prose and Table 3, there are reports of language, functional communication, and other psychosocial assessments, and their relevance is not made clear to the reader
Results section 3.2 is very interesting and the data on these 4 studies is clearly summarised here and the three phase approach is a useful one for conceptualising involvement.
It is not clear how results section 3.5 is based on data derived from the studies; it appears instead to be prose which relates more to Discussion. Authors are encouraged to either make the distinction clear and retain as results, or incorporate into the Discussion.
Discussion
Line 151: it’s not clear how authors have determined the 98% of studies. Indeed, the paper appears to report involvement in 4/20 studies, so this figure should be lower at 75%
Line 166: the authors note the lack of reporting guidance regarding PPI. Whilst in general I agree with the tenet that reporting has been poorly done in aphasia research literature over the years, this specific claim is not completely true - there are now internationally accepted reporting guidelines and these were published recently in 2017, which could be applied to reporting on PWA involvement in aphasia research. See GRIPP2 https://www.equator-network.org/reporting-guidelines/gripp2-reporting-checklists-tools-to-improve-reporting-of-patient-and-public-involvement-in-research/. It would be important to see authors drawing on these guidelines to their discussion and reflecting on how their results relate to these guidelines. A fuller consideration would be to revisit the studies in the light of these guidelines, however I appreciate that this is a considerable undertaking, and the framework already used in this paper (three phases) is a viable approach
The discussion champions involvement and advocates for involvement of PWA, however the reader wonders how the findings compare to other fields, and whether indeed this is a problem solely for aphasiology, or whether it is a broader problem across other fields (stroke, dementia, etc) which perhaps deepens the argument. Ideally the authors would also use research to demonstrate how involvement of users in development of quality of life questionnaires has altered the endpoint outcome, thereby demonstrating the real value of why this is so important.
The authors need to more fully reflect on the limitations of this study specifically the constrained date searching, and more importantly the challenge inherent in making claims based on reporting in studies (see point made in opening paragraph) at minimum.
Author Response
|
Reviewer’s Comment |
Authors Response |
|
Firstly, the authors need to acknowledge that using research articles as a means for assessing involvement is reliant on the reporting of that involvement in the said articles – so it is not possible on this basis to claim that various quality of life tools or aphasia-impact-related tools have not involved people with aphasia in their development. This is because reporting (of many and varied aspects of the research process) is often incomplete or entirely omitted at the publication stage; this does not mean it did not happen |
Thank you for this important statement, which will clarify the methodology.
The authors will clarify in the manuscript that the involvement of PWA in QOL and AIR was analyzed and presented as reported in the selected published studies. Authors added the sentence 200-201 in page 6. Since there was no mention by the researchers of the involvement of PWA in these studies, the present authors made the natural assumption that PWA were not involved.
However, based on the reviewer’s comments, it is clear that researchers need to be more transparent in their methodology and implicitly state the involvement of PWA, and in what capacity. That this kind of information is omitted at the publication stage, is a serious flaw for the facilitation and interpretation of PPI. |
|
For example, in this paper, the authors state that the original development of the SAQOL-39 did not involve stroke survivors with aphasia based on their reading of reference #37 (2003). I am not the author of this QOL tool, but I know it well and I know this claim to be inaccurate regarding the tool. An earlier paper (2001)https://pubmed.ncbi.nlm.nih.gov/11340850/ highlights two different ways people with aphasia were involved, and indeed, the SSQOL on which the SAQOL was developed initially did involve stroke survivors which was one of the main reasons that Hilari selected the SSQOL for adaptation as her PhD. As such, it is essential that the authors more carefully explain their data is derived from reference to specific research articles, rather than making claims about the tool itself. |
Given this important remark, we went again to the paper on SAQOL by Hilary et al (2013) #37. They did not mention involving PWA in the creation of the questionnaire as stated in the published study.
There was no intention from the authors to mislead the readers or misrepresent information, but only to state as is, what was reported by Hilari herself. Information presented in the current study was strictly based on the published studies of the 10-year period mentioned in the methodology.
We also looked at the original 2001 Paper and found the following sentence: “People with aphasia were not included in the process of developing the SS-QOL, and this has implications on its content validity with thisgroup of people” p.88. of SSQoL published paper by Hilari and Byng1 (2001). Nevertheless, SS-QOL was pilot tested with 12 PWA and 18 people with chronic aphasia volunteered to participate in the field pre-test of the tool.
1Hilari, Katerina & Byng, Sally. (2001). Measuring Quality of Life in People with Aphasia: The Stroke Specific Quality of Life Scale. International journal of language & communication disorders / Royal College of Speech & Language Therapists. 36 Suppl. 86-91. 10.3109/13682820109177864. We will clarify this in a sentence.
|
|
Secondly, the authors need to provide a credible rationale for why the January 2010 start date for their search; there is no viable reason that exists in the field, and indeed QOL tools have been in development since 2000. Had their search started earlier, they would have found the 2001 publication above in their search.
|
This work is based on scoping review methods and not a systematic review ref#22 (Arksey, H.; O’Malley, L. Scoping studies: Towards a methodological framework. Int J Soc Res Methodol Theory Pract 2005, 8, 19–32.). Automatically this framework refers to the investigation of the later publications in the proposed field and not every possible study conducted in the investigated area. A search spanning the last decade is the common process used in scoping review methodology in health sciences. Scoping reviews ‘aim to map rapidly the key concepts underpinning a research area and the main sources and types of evidence available, and can be undertaken as stand-alone projects in their own right, especially where an area is complex or has not been reviewed comprehensively before’ (Mays, Roberts, & Popay, 2001, p. 194). Scoping reviews tend to search the last 10 years in the selected area of research as they search the latest information, trends, knowledge gapson the proposed topics. Scoping reviews aim to raise new research questions in currentbibliography and not in bibliography in general. Please see the scoping review by Conklin et al., (2012) 1 on PPI and health-care policy with a 10 year time span, from November 2000 to April 2010, the scoping review by Krishnan et al., (2017) 2 on Stroke Care and Caregivers perspectives, where again they had a ten year time span (2003 until 2014), the scoping review by Griffel et al., (2020)3 who conducted a ten year search (2009–2019) on patient-center therapy apps and many others. 1Conklin, Annalijn & Morris, Zoë & Nolte, Ellen. (2012). What is the evidence base for public involvement in health-care policy?: Results of a systematic scoping review. Health expectations : an international journal of public participation in health care and health policy. 18. 10.1111/hex.12038. 2Krishnan, Shilpa & Pappadis, Monique & Weller, Susan & Stearnes, Marsja & Kumar, Amit & Ottenbacher, Kenneth & Reistetter, Timothy. (2017). Needs of Stroke Survivors as Perceived by Their Caregivers: A Scoping Review. American Journal of Physical Medicine & Rehabilitation. 1. 10.1097/PHM.0000000000000717. 3Griffel, Jenny & Leinweber, Juliane & Spelter, Bianca & Roddam,. (2020). Patient-centred design of aphasia therapy apps: a scoping review.
|
|
Additionally, authors are advised to consider the reporting standards now available for PPI https://www.equator-network.org/reporting-guidelines/gripp2-reporting-checklists-tools-to-improve-reporting-of-patient-and-public-involvement-in-research/ |
The authors are effectively focusing on PPI reporting standards in PWA. This information was missing from the selected studies and as a result, could not being used in aphasia research protocols during the last decade. We have mentioned this point in the Introduction, lines 72-75. |
|
The introduction takes a strong exclusion stance, which is needed to create the argument for this research. However, it does not reflect at least a decade of research in aphasia rehabilitation research where researchers have focused almost entirely on clients’ or users’ views (e.g. Kyla Brown’s research on living successfully with aphasia; Sarah Wallace’s research on prioritised outcomes of rehabilitation research; Molly Manning’s very recent review). It is slightly disingenuous not to consider this. |
Thank you for this precious point. Line 151: “Title screening resulted in the removal of 767 additional papers, where titles did not concern QoL and/or AIR-Qs but instead were related to aphasia rehabilitation”.
It is clearly stated in the article that the intention of the authors was not to include studies related to aphasia rehabilitation but only on studies including QOL and AIR tools. |
|
Research question 3 is also missing
|
Question 3 is stated in line 119 and its analyzed in Section 3 (Results) under point 3.3. page 16 line 104. |
|
So Table 2 reports on gold standard measures which relates to psychometrics comparisons that the papers would have reported on if psychometrics were a focus of the paper, so it’s unclear why this relates to PPI reporting in this study. Similarly, in the prose and Table 3, there are reports of language, functional communication, and other psychosocial assessments, and their relevance is not made clear to the reader
|
|
|
Line 151: it’s not clear how authors have determined the 98% of studies. Indeed, the paper appears to report involvement in 4/20 studies, so this figure should be lower at 75%
|
Thank you for the correction of the percentage. Yes 75% is the correct number, we already changed in the article, line 226. |
|
Line 166: the authors note the lack of reporting guidance regarding PPI. Whilst in general I agree with the tenet that reporting has been poorly done in aphasia research literature over the years, this specific claim is not completely true - there are now internationally accepted reporting guidelines and these were published recently in 2017, which could be applied to reporting on PWA involvement in aphasia research. See GRIPP2 https://www.equator-network.org/reporting-guidelines/gripp2-reporting-checklists-tools-to-improve-reporting-of-patient-and-public-involvement-in-research/.
|
Thank you for this clarification. Authors have included the GRIPP2 framework in lines 150-151- besides the addition in the introduction.
Again, as mentioned above, the authors are not discussing general PPI reporting frameworks in Health Sciences but are focusing on the absence of PPI reporting in aphasia research protocols, and specifically in the selected studies where PWA are reported by the authors to be involved as research partners, stakeholders, co researchers, co-producers etc.
|
|
It would be important to see authors drawing on these guidelines to their discussion and reflecting on how their results relate to these guidelines. A fuller consideration would be to revisit the studies in the light of these guidelines, however I appreciate that this is a considerable undertaking, and the framework already used in this paper (three phases) is a viable approach.
|
This is a good suggestion by the reviewer, and the proposed approach can be implemented in a new research study.
One recommendation from the authors will be that researchers who are translating, adapting and validating pre-established QOL and AIR tools into other languages should involve PWA in their study protocols and as research partners throughout the study using the GRIPP2 reporting checklist or any other relevant framework, line 325-326.
Ref #69 Staniszewska, S.; Brett, J.; Simera, I.; Seers, K.; Mockford, C.; Goodlad, S.; et al. GRIPP2 reporting checklists: Tools to improve reporting of patient and public involvement in research. BMJ Online 2017, 358, doi.org/10.1136/bmj.j3453
|
|
The discussion champions involvement and advocates for involvement of PWA, however the reader wonders how the findings compare to other fields, and whether indeed this is a problem solely for aphasiology, or whether it is a broader problem across other fields (stroke, dementia, etc) which perhaps deepens the argument.
|
Although the authors acknowledge, given also their long clinical experience, that patients are highly considered when constructing such tools, this seems not to be formally reported. A brief review of the literature on QoL constructs for patients with cancer (Coens et al., 2020) reveal that people with cancer were not reported to be co-creators or consultants on the creation of the QoL questionnaires regarding living with cancer. It appears researchers either fail to take their involvement as being necessary or like the reviewer suggests (for aphasia) may have asked people with cancer to participate in but fail to report so. So, this does not appear to be a problem solely for aphasiology but a broader problem when methodology is not transparent.
Coens, Corneel et al. “International standards for the analysis of quality-of-life and patient-reported outcome endpoints in cancer randomised controlled trials: recommendations of the SISAQOL Consortium.” The Lancet. Oncology vol. 21,2 (2020): e83-e96. doi:10.1016/S1470-2045(19)30790-9
|
|
Ideally the authors would also use research to demonstrate how involvement of users in development of quality of life questionnaires has altered the endpoint outcome, thereby demonstrating the real value of why this is so important.
|
Reviewer 2 Report
I found this metaanalysis convincing, in terms of the criteria for metaanalysis, dsplaying of results, and conclusions drawn therefrom.
My main criticisms are:
- The number of abbreviations is overwhelming. I do appreciate the addition of a glossary, but it does not appear until the end of the paper, causing the reader to make many cross-checks.
- I think the authors' expectation that patients with aphasia should be involved in every aspect of a study, from conception and methodology to data analysis, conclusions, and significance is a bit of a high bar. As they mention, some patients with mild aphasia who are otherwise highly educated and informed might be capable of this, most would not. In addition, I am not sure that other handicapped individuals have been expected to play this extensive a role in research on their condition.
Author Response
|
Reviewers comment |
Authors Response |
|
The number of abbreviations is overwhelming. I do appreciate the addition of a glossary, but it does not appear until the end of the paper, causing the reader to make many cross-checks.
|
Thank you for this comment. Indeed, there are a lot of abbreviations in the article but still needed in order to follow the structure of the tools. After your recommendation, authors have removed the abbreviations that are only used once and there is no need to refer back to them e.g. in line 284 the American Speech and Hearing Association- Quality of Communication Life Scale- ASHA QCLS and the Royal College of Speech and Language Therapists -RCSLT in line 290-291. Hopefully the glossary will help the reader follow necessary abbreviations. |
|
I think the authors' expectation that patients with aphasia should be involved in every aspect of a study, from conception and methodology to data analysis, conclusions, and significance is a bit of a high bar. As they mention, some patients with mild aphasia who are otherwise highly educated and informed might be capable of this, most would not. In addition, I am not sure that other handicapped individuals have been expected to play this extensive a role in research on their condition.
|
This is an excellent comment and thank you for your input. Indeed, people with severe aphasia would find it challenging to respond and engage in all aspects of a study. Nevertheless, authors suggest that the involvement of PWA must be adapted to the severity of the aphasia (Sten and Wagner position paper in line 270) and to their cognitive reserve. Based on the reviewer’s comment the authors also added a line about the addition of the communication partners of PWA in research protocols in page 20, line 328. |
Round 2
Reviewer 1 Report
The authors have responded to comments adequately.